# Post-Surgical Depositions of Blood Products Are No Major Confounder for the Diagnostic and Prognostic Performance of CEST MRI in Patients with Glioma

**DOI:** 10.3390/biomedicines11092348

**Published:** 2023-08-23

**Authors:** Nikolaus von Knebel Doeberitz, Florian Kroh, Laila König, Philip S. Boyd, Svenja Graß, Cora Bauspieß, Moritz Scherer, Andreas Unterberg, Martin Bendszus, Wolfgang Wick, Peter Bachert, Jürgen Debus, Mark E. Ladd, Heinz-Peter Schlemmer, Steffen Goerke, Andreas Korzowski, Daniel Paech

**Affiliations:** 1Division of Radiology, German Cancer Research Center (DKFZ), 69120 Heidelberg, Germany; 2Division of Medical Physics in Radiology, German Cancer Research Center (DKFZ), 69120 Heidelberg, Germany; 3Department of Physics and Astronomy, University of Heidelberg, 69120 Heidelberg, Germany; 4Department of Radiation Oncology, Heidelberg University Hospital, 69120 Heidelberg, Germany; 5Department of Neurosurgery, University Hospital Heidelberg, 69120 Heidelberg, Germany; 6Faculty of Medicine, University of Heidelberg, 69120 Heidelberg, Germany; 7Department of Neuroradiology, University Hospital Heidelberg, 69120 Heidelberg, Germany; 8Department of Neurology, University Hospital Heidelberg, 69120 Heidelberg, Germany; 9Clinical Cooperation Unit Radiation Oncology, German Cancer Research Center (DKFZ), 69120 Heidelberg, Germany; 10Department of Neuroradiology, University Hospital Bonn, 53127 Bonn, Germany

**Keywords:** chemical exchange saturation transfer MRI, amide proton transfer, semi-solid magnetization transfer, glioma, radiotherapy, therapy response, overall survival, blood, correction, hemosiderin, methemoglobin

## Abstract

Amide proton transfer (APT) and semi-solid magnetization transfer (ssMT) imaging can predict clinical outcomes in patients with glioma. However, the treatment of brain tumors is accompanied by the deposition of blood products within the tumor area in most cases. For this reason, the objective was to assess whether the diagnostic interpretation of the APT and ssMT is affected by methemoglobin (mHb) and hemosiderin (Hs) depositions at the first follow-up MRI 4 to 6 weeks after the completion of radiotherapy. A total of 34 participants underwent APT and ssMT imaging by applying reconstruction methods described by Zhou et al. (APTw_asym_), Goerke et al. (MTR_Rex_APT and MTR_Rex_MT) and Mehrabian et al. (MT_const_). Contrast-enhancing tumor (CE), whole tumor (WT), mHb and Hs were segmented on contrast-enhanced T_1_w_CE_, T_2_w-FLAIR, T_1_w and T_2_*w images. ROC-analysis, Kaplan–Meier analysis and the log rank test were used to test for the association of mean contrast values with therapy response and overall survival (OS) before (WT and CE) and after correcting tumor volumes for mHb and Hs (CE_C_ and WT_C_). CE_C_ showed higher associations of the MTR_Rex_MT with therapy response (CE: AUC = 0.677, *p* = 0.081; CE_C_: AUC = 0.705, *p* = 0.044) and of the APTw_asym_ with OS (CE: HR = 2.634, *p* = 0.040; CE_C_: HR = 2.240, *p* = 0.095). In contrast, WT_C_ showed a lower association of the APTw_asym_ with survival (WT: HR = 2.304, *p* = 0.0849; WT_C_: HR = 2.990, *p* = 0.020). Overall, a sophisticated correction for blood products did not substantially influence the clinical performance of APT and ssMT imaging in patients with glioma early after radiotherapy.

## 1. Introduction

Standard of care for diffuse glioma includes maximum safe resection, with subsequent radio- and chemotherapy [1]. Yet, since treatment-related changes, such as pseudoprogression and radionecrosis have similar morphological imaging features compared to progressing glioma tissue, therapy response assessment and survival prediction are well-known challenges in clinical neuro-oncology, with possibly harmful consequences for the patient [2]. In this regard, the promising clinical potential of functional MR imaging methods, including perfusion-weighted imaging, MR spectroscopy and chemical exchange saturation transfer (CEST) imaging has been shown in recent years [3,4]. Imaging of the amide-proton transfer (APT) and of the semi-solid magnetization transfer (ssMT) relies on the selective radio frequency (RF) saturation of protons bound in peptide bindings or sub-cellular macromolecules, with subsequent magnetization transfer to bulk water through chemical exchange or spin–spin couplings, respectively [5] (collectively referred to as CEST imaging in the following). Several groups have demonstrated that APT and ssMT imaging could predict the therapy response and survival of patients with glioma before and after radiochemotherapy [6,7,8,9,10]. Others have demonstrated that imaging of the APT and ssMT could also be used to differentiate radiation-induced changes from tumor progression as early as the first follow-up after the completion of radiotherapy [11,12,13,14,15,16,17]. However, treatment of brain tumors is frequently accompanied by perioperative and radiation-induced disruptions of the blood–brain-barrier with subsequent depositions of blood products, such as methemoglobin (mHb) and hemosiderin (Hs), in the tumor area [18]. mHb and Hs are both proteinaceous, contain paramagnetic iron (Fe^3+^) and exist to varying degrees as conglomerates of insoluble macromolecules. For this reason, mHb and Hs not only have a strong influential impact on T_1_ and T_2_, but should also contribute to the APT and ssMT pool. However, whilst several groups have demonstrated that CEST imaging of the APT can differentiate between acute and subacute stages of intracranial hemorrhage [19,20,21], little is known about the influence of mHb and Hs on the clinical performance of APT and ssMT imaging in patients with glioma in the post-radiotherapy interval. Furthermore, CEST contrasts are heavily dependent on the metrics used for their reconstruction form the Z-spectrum [7,22,23]. Therefore, the purpose of this study was to assess the relevance of advanced correction methods for mHb and Hs depositions in the tumor area for the clinical performance of APT and ssMT contrasts under the application of different reconstructions methods first described by Zhou et al. (APTw_asym_) [24], Goerke et al. (MTR_Rex_APT and MTR_Rex_MT) [25] and Mehrabian et al. (MT_const_) [7].

## 2. Materials and Methods

Eligible for this prospective clinical study were all patients who received radiotherapy for diffuse glioma at the Department of Radiation-Oncology of the University Hospital Heidelberg between September 2018 and December 2021, and who were 18 years of age or older, had a Karnofsky Performance Score of at least 50 and had the legal capacity to consent. Eventually, 72 study participants (61 with initial disease and 11 with relapsing/progressive disease) were enrolled and received CEST imaging at the first follow-up MRI 4 to 6 weeks after the completion of radiotherapy. Two participants had to be excluded from the analysis due to heavy motion artifacts, seven due to incomplete datasets and one for excessive perioperative ischemia. The data cut-off was 3 May 2022. The association of CEST imaging with therapy response and progression-free survival (PFS), as well as with overall survival (OS), was previously investigated in two studies involving 61 and 49 participants of the same cohort, respectively [5,10]. Therapy response was assessed based on longitudinal clinical data and MRI according to the revised response assessment in neuro oncology (RANO) criteria by two radiologists with 6 (N.v.K.D.) and 11 (D.P.) years of experience in neuroimaging at the time of data acquisition [26]. The results of the assessment were reconciled with institutional multidisciplinary tumor board decisions to account for potential changes in relevant medications, such as antiangiogenic or cytotoxic drugs, and changes in clinical status. Overall survival (OS) was assessed by written request to the relevant public registries and was available for 54 of 62 evaluable participants. Due to differences in the tumor biology between midline gliomas and hemispherical gliomas, the data of five study participants with midline gliomas were additionally excluded from the analysis [27,28]. Finally, given that the associations of the investigated contrasts with therapy response and survival are influenced by the presence of residual contrast enhancement on MRI [6,11], the data of 34 participants with available survival data, hemispherical gliomas and residual contrast enhancement were analyzed (Figure 1).

Histology: For all of the 34 study participants evaluated, tumor tissue was available for histopathological analysis after biopsy or surgical resection. Routinely, *IDH*-, *ATRX*-, LOH1p19q- and *MGMT*-status were assessed, and histopathological classification was performed in accordance with the 2016 version of the World Health Organization (WHO)’s criteria for the classification of primary central nervous system (CNS) tumors. Please see Table 1 for a detailed description of the histopathological tumor characteristics for all evaluated participants.

Image Acquisition and Postprocessing: Image acquisition was performed on a 3T whole-body MR scanner (MAGNETOM Prisma; Siemens Healthineers, Erlangen, Germany) with an integrated transmit body coil and a 64-channel head/neck receiving coil. The CEST data were processed in Matlab^®^ (Mathworks, version 2019b, Natick, MA, USA) using customized scripts.

Imaging of the APT and ssMT according to Goerke et al. (MTR_Rex_APT and MTR_Rex_MT): A 3D spiral-centrally reordered gradient-echo acquisition sequence (snapshot CEST [29,30]) was applied with the same image readout parameters (matrix = 128 × 104 × 16, resolution = 1.7 × 1.7 × 3 mm^3^) and presaturation as previously described by Goerke et al. [25]. For presaturation, trains of 148 Gaussian-shaped radio frequency (RF) pulses (echo time (TE) = 2.75 ms, repetition time (TR) = 5.5. ms, flip angle = 7°, puls length (tp) = 0.02 s and duty cycle = 80%) with two amplitudes (B_1_ = flip angle/(γ·tp)) of 0.6 μT and 0.9 μT were acquired at 57 unequally distributed offsets in the range between ±250 ppm and −300 ppm for normalization at two M_0_, resulting in a saturation time of 3.7 s and a total measurement time of 7:34 min. The WASABI [31] (3:41 min) approach was applied to yield B_0_ and B_1_ maps, using the same image readout and similar presaturation parameters as described above. In this case, presaturation was performed by sampling 31 equally distributed frequency offsets around ±2 ppm. For post-processing, the CEST and WASABI data were first co-registered with a rigid registration algorithm in MITK (version v2022.10). Then, the CEST data were processed in Matlab^®^ (Mathworks, version 2019b, Natick, MA, USA). A correction of B_0_ inhomogeneities was achieved by shifting the Z-spectra along Δω [32] and denoising was achieved under the application of a principle component-based algorithm [32]. The reconstruction of the MTR_Rex_APT and MTR_Rex_MT from the Z-spectrum was performed as described in [25] under the application of a four-pool Lorentzian-fit ±250 ppm up- and down-field of the water offset (0 ppm: direct water saturation, 3.5 ppm: APT, −3.5 ppm: rNOE, and −2.5 ppm: ssMT) with $MTR_{Rex}=\frac{1}{Z}-\frac{1}{Z_{ref}}$ (Z = all fitted pools, Z_ref_ = fitted pools—pool of interest). A two-point “contrast-correction” method first proposed by Windschuh et al. [33] was applied for additional B_1_-correction.

Imaging of the ssMT according to Mehrabian et al. (MT_const_): Image readout parameters, presaturation, co-registration of CEST data and B_0_ correction were the same as described above. The MT_const_ was reconstructed from the Z-spectrum of the CEST data with B_1_ = 0.6 µT with a Lorentzian fit around ±6 ppm with [7]:$$S\left(\Delta \right)=1-\left(MT+\sum_{i=1}^{4}\frac{A_{i}}{1+{\left(\frac{\Delta -\Delta_{0i}}{0.5w_{i}}\right)}^{2}}\right)$$
(1)$$\left[A_{i},\Delta_{0i},w_{i}\right]=[amplitude,centrefrequency,width]$$

Imaging of the APT according to Zhou et al. (APTw_asym_): Again, pulse sequence and image readout parameters were the same as described above [24]. In accordance with recent consensus guidelines [23], four rectangular RF pulses with a B_1_ of 2 µT (t_sat_ = 0.2 s and 95% duty cycles) at 16 frequency offsets at ±4 (1), ±3.75 (2), ±3.5 (2), ±3.25 (2), and ±3 (1) ppm and an additional M_0_ at −300 ppm were obtained, resulting in a scan time of 2:00 min. Co-registration of the CEST data and B_0_-correction of the Z-spectra was performed under the application of similar post-processing methods as described above with APTw = Z(−3.5 ppm) − Z(3.5 ppm).

Quantitative T_1_ mapping: The longitudinal relaxation time of water (T_1_) was measured via quantitative mapping with the same image readout parameters as above. A saturation recovery sequence with recovery times (t_rec_) of 0.1, 0.25, 0.5, 1.0, 1.5, 2.0, 2.5, 3.5, 5.0, 7.5 and 10.0 s and $M_{z}\left(t_{rec}\right)=M_{0}+(M_{z}\left(0\right)-M_{0})\cdot e^{-t_{rec}/T_{1}}$ [25] was applied, resulting in a scan time of 1:15 min.

Segmentation of tumor volumes and blood products: Three-dimensional segmentations of tumor volumes and blood products were also performed in Matlab^®^ (Mathworks, version 2019b, Natick, MA, USA). mHb (detectable in 14 cases) was defined as hyperintense material on T_1_w, and Hs (detectable in 33 cases) as dark tissue on T_2_*w susceptibility-weighted imaging (SWI). Contrast-enhancing (CE) and whole tumor volumes (WT) were segmented on contrast-enhanced T_1_w (T1w_CE_) and T_2_w fluid-attenuated inversion recovery (T_2_w-FLAIR) images. WT encompassed CE plus adjacent T_2_w-FLAIR-hyperintense signal alterations. Larger mHB depositions that were identifiable as such on T_1_w_CE_ and T_2_w-FLAIR (e.g., due to localization in the resection cavity or associated T_2_w-FLAIR signal drops that indicated surrounding Hs (Figure 2b)) were grossly excluded (Appendix A). Contrast-enhancing and whole tumor volumes corrected for mHb and Hs (CE_C_ and WT_C_) were calculated from CE and WT in Matlab^®^ by subtracting the overlapping mHb and Hs volumes (Figure A1).

Statistical analyses: Mann–Whitney-U-test was applied to test for differences between the mean CEST contrast values of blood products and corrected tumor volumes, as well as between uncorrected and corrected tumor volumes. Receiver operating characteristic (ROC) analyses were performed to test for the association of mean CEST contrast values of uncorrected and corrected tumor volumes with therapy response, as assessed according to the RANO criteria. Kaplan–Meier analyses and log rank tests were used to test for the association of mean CEST contrast values of uncorrected and corrected tumor volumes with OS. In-house software in Matlab^®^ (Mathworks, version 2019b, Natick, MA, USA) was used for all statistical analyses. *p* ≤ 0.05 was considered as being statistical significant.

## 3. Results

In total, 72 study participants (mean age 59 ± 16 years; 43 male) underwent CEST MRI at the first follow-up 4 to 6 weeks after the completion of radiotherapy. The data of 34 participants (19 male, 15 female; mean age of 59.2 ± 15.6 years) with diffuse hemispherical glioma who had available data on therapy response and survival, and showed residual contrast enhancement on MRI were included in the analysis. A total of 16 participants were assessed as having stable disease (SD) and 18 were assessed as having progressive disease (PD). Median OS was 287 days (min. 63 and max. 1271 days), with 24/34 participants having reached an endpoint by the data cut-off on 3 May 2022. Detailed clinical characteristics of the analyzed study cohort are provided in Table 1.

### 3.1. CEST Contrast Maps of Participants with Larger Depositions of mHb and Hs

Exemplary contrast maps of four study participants with larger depositions of mHb (2) and Hs (2) in the tumor area are displayed in Figure 2 and Figure 3, respectively. Associated quantitative T_1_ maps are depicted in Figure A2. Visually, larger depositions of Hs showed remarkably higher values on the MTR_Rex_APT contrast maps sharply confined to the drawn ROIs (Figure 3). This was not observed on the other contrast maps. Furthermore, larger mHb depositions visually showed remarkably dropped values on MTR_Rex_APT, MTR_Rex_MT, MT_const_ and T_1_ (Figure 2 and Figure A2) contrast maps, which was recapitulated by lower MTR_Rex_APT, MTR_Rex_MT and MT_const_ mean values of mHb in comparison to CE_C_ and WT_C_ (Table A1). Exemplary fitted Z-spectra for representative voxels of contrast-enhancing tumor tissue, peritumoral T_2_w-FLAIR-hyperintense signal alterations, and mHb and Hs in an exemplary participant with larger depositions of mHb and Hs are shown in Appendix A.

### 3.2. Differences between the Mean CEST Contrast Values of Uncorrected and Corrected Tumor Volumes

Even though there were differences between the mean MTR_Rex_APT, MTR_Rex_MT and MT_const_ contrast values of mHb and CE_C_/WT_C_, as well as the mean MTR_Rex_MT and APTw_asym_ contrast values of Hs and CE_C_/WT_C_ (Table A1), there were no differences between the mean values of any contrast for uncorrected and corrected tumor volumes (Figure 4). The mean MTR_Rex_APT values were 0.249 ± 0.036 vs. 0.247 ± 0.037 (*p* = 0.854) for CE vs. CE_C_, and 0.243 ± 0.029 vs. 0.243 ± 0.029 (0.990) for WT vs. WT_C_, respectively. For the MTR_Rex_MT the mean contrast values were 0.376 ± 0.071 vs. 0.381 ± 0.073 (*p* = 0.695) for CE vs. CE_C_, and 0.464 ± 0.065 vs. 0.470 ± 0.063 (*p* = 0.615) for WT vs. WT_C_, respectively. For the APTw_asym_ the mean contrast values were 1.388 ± 0.563% vs. 1.379 ± 0.553% (*p* = 0.893) for CE vs. CE_C_, and 0.914 ± 0.536 vs. 0.898 ± 0.528% (*p* = 0.759), respectively. The mean contrast values of the MT_const_ for CE vs. CE_C_ were 0.171 ± 0.029 vs. 0.173 ± 0.028 (*p* = 0.704), and those for WT vs. WT_C_ were 0.162 ± 0.023 vs. 0.163 ± 0.023 (*p* = 0.023).

### 3.3. Association of CEST Contrast Values of Uncorrected and Corrected Tumor Volumes with Therapy Response

In the ROC analyses, the MTR_Rex_MT was the only contrast that showed a noticeable improvement regarding the association of the mean contrast values of tumor tissue with therapy response at the first follow-up. The area under the curve (AUC) for differentiating participants with PD and SD according to the mean contrast values (with PD > SD) was 0.677 (*p* = 0.081) for CE and 0.705 (*p* = 0.044) for CE_C_. However, there were no differences in AUCs for WT (AUC = 0.635, *p* = 0.184) and WT_C_ (AUC = 0.628, *p* = 0.184). The association of MT_const_ mean contrast values with therapy response of uncorrected and corrected tumor volumes did not show relevant differences (CE: AUC = 0.826, *p* = 0.001; CE_C_: AUC = 0.816, *p* = 0.002; WT: AUC = 0.868, *p* < 0.001; WT_C_: AUC = 0.861, *p* < 0.001). The MTR_Rex_APT (CE: AUC = 0.438, *p* = 0.546; CE_C_: AUC = 0.424, *p* = 0.458; WT: AUC = 0.566, *p* = 0.523; WT_C_: 0.569, *p* = 0.501) and APTw_asym_ (CE: AUC = 0.514, *p* = 0.904; CE_C_: AUC = 0.504, *p* = 0.986; WT: AUC = 0.538, *p* = 0.717; WT_C_: AUC = 0.552, *p* = 0.617) mean contrast values did not show any association with therapy response before and after the correction. The ROC curves for the investigated CEST contrasts are displayed in Figure 5. The results of the ROC analysis are summarized in Table A2.

### 3.4. Association of Mean CEST Contrast Values of Uncorrected and Corrected Tumor Volumes with Overall Survival

In the Kaplan–Meier analyses, the APTw_asym_ mean contrast values showed a slightly lower association with survival for CE_C_ (CE: HR = 2.634, *p* = 0.040; CE_C_: HR = 2.634, *p* = 0.040) and a slightly higher association with survival for WT_C_ (WT: HR = 2.304, *p* = 0.084; WT_C_: HR = 2.990, *p* = 0.020), compared to the respective uncorrected tumor volumes, with a shorter OS of participants with higher mean values compared to the cohort median (CE: 215 vs. 392 days; CE_C_ 215 vs. 392 days; WT: 225 vs. 392 days; WT_C_ 215 vs. 392 days). The MTR_Rex_APT mean values also showed a lower association with survival for CE_C_ (CE: HR = 2.439, *p* = 0.056, OS = 225 vs. 416 days; CE_C_: HR = 2.110, *p* = 0.110, OS = 253 vs. 392 days), whilst for whole tumor volumes, no association with survival could be observed regardless of the correction (WT: HR = 1.526, *p* = 0.417, OS = 225 vs. 392 days; WT_C_: HR = 1.525, *p* = 0.417, OS = 225 vs. 392). For the MT_const_, the correction had no measurable impact on its trend towards an association with survival for contrast-enhancing tumor volumes (CE: HR = 2.330, *p* = 0.068, OS = 228 vs. 315 days; CE_C_: HR = 2.330, *p* = 0.068, OS = 228 vs. 315 days) and on its association with survival for whole tumor volumes (WT: HR = 2.536, *p* = 0.044, OS = 215 vs. 392 days; WT_C_: HR = 2.535, *p* = 0.044, OS = 215 vs. 392 days). The MTR_Rex_MT was not associated with survival regardless of the correction (CE: HR = 0.958, *p* = 0.919, OS = 315 vs. 280 days; CE_C_: HR = 1.068, *p* = 0.964, OS = 315 vs. 280 days; WT: HR = 1.389, *p* = 0.559, OS = 294 vs. 280 days; WT_C_: HR = 1.179, *p* = 0.847, OS = 315 vs. 225 days). The Kaplan–Meier plots are depicted in Figure 6 and the results are summarized in Table A3.

### 3.5. Supermedian Analysis of the Mean CEST Contrast Values of Uncorrected and Corrected Tumor Volumes

To understand the impact of the blood product correction on the tumor-associated mean CEST contrast values in greater detail, we also assessed how many participants switched from the respective groups with higher mean CEST contrast values compared to the respective cohort medians (supermedian) to the groups with lower mean values (submedian) and vice versa upon correcting tumor volumes for mHb and Hs. We observed that for CE_C_ in comparison to CE, one participant switched from super- to submedian and one participant from sub- to supermedian for MTR_Rex_APT, MTR_Rex_MT and APTw_asym_. Additionally, for WT_C_ in comparison to WT one participant switched from super- to submedian and one participant from sub- to supermedian for MTR_Rex_MT. However, for the MT_const_, no participants switched from super- to sub- or sub- to supermedian upon correcting any tumor volume for blood products. The results of this analysis with corresponding tumor mean CEST contrast values for the assessed participants and respective cohort medians are summarized in Appendix A.

## 4. Discussion

Whilst several groups demonstrated the potential of CEST imaging in the diagnostic follow-up after hemorrhagic stroke, little is known about the influence of post-therapeutic depositions of blood breakdown products on the clinical performance of APT and ssMT imaging in the early post-radiotherapy interval. For this reason, the purpose of this study was to assess the impact of advanced correction methods for mHb and Hs on the association of most commonly employed APT and ssMT contrasts with therapy response and OS at the first follow-up 4 to 6 weeks after the completion of radiotherapy at 3T. Even though, the MTR_Rex_APT contrast maps showed markedly elevated values in correspondence with Hs, and the MTR_Rex_APT, MTR_Rex_MT and MT_const_ contrast maps showed noticeably dropped values in correspondence with mHb, no relevant differences in mean contrast values between uncorrected and corrected tumor volumes could be detected. However, for corrected contrast-enhancing tumor volumes, a slightly stronger association of the MTR_Rex_MT with therapy response was observed, whilst the MTR_Rex_APT showed a moderately weaker association with survival. Interestingly, the APTw_asym_ showed contradicting trends, with a somewhat weaker association with survival for corrected contrast-enhancing tumor volumes, and a slightly stronger association with survival for corrected whole tumor volumes. Concurrently, the association of MT_const_ mean values with therapy response and survival were unaffected by the correction.

In previous studies, Sawaya et al., Wang et al. and Ma et al. observed that asymmetry-based APTw imaging showed markedly elevated contrast values in rat models and in patients with acute and subacute cerebral bleeding, which very likely corresponded to accumulations of deoxygenized hemoglobin and mHb [13,19,20,34,35,36]. Lai et al., on the other hand, observed significantly reduced contrast values in the subacute stage of cerebral hemorrhage in a preclinical study, using an apparent exchange-dependent relaxation compensated metric of the APT (APT_AREX_) [21]. Contrary to these findings, to our knowledge, there are no available published results on the CEST contrast behaviors of Hs.

The findings of Lai et al. are mirrored by decreased MTR_Rex_APT, MTR_Rex_MT and MT_const_ values for mHb in comparison to corrected tumor volumes, which were observed on this study (Figure 2). Given that mHb contains paramagnetic Fe^3+^, the observed contrast patterns might at least in part be explained by residual T_1_ contributions especially to the MT_const_, but to a lesser extent also to the other investigated CEST contrasts [5,6,7,22,34]. Since magnetization transfer between protons and free water through chemical exchange is base-triggered, pH might be another factor that could influence particularly APT-weighted CEST contrasts [5,22]. The visibly increased values on MTR_Rex_APT contrast maps that corresponded to Hs, on the other hand, are harder to explain (Figure 2). Hs consists of intracellular accumulations of insoluble and partially digested ferritin, which should be associated with a rather acidotic intralysosomal milieu, T_1_ contributions from paramagnetic Fe^3+^ and fewer mobile amide protons due to the insoluble state of the proteinaceous compounds [34]. Taken together, further research is needed to understand the physico-chemistry behind the observed CEST contrast patterns of mHb and Hs.

Despite these observations and considerations, no relevant differences in the investigated CEST contrast mean values could be observed between the uncorrected and corrected tumor volumes. Concurrently, the correction only had a very minor impact on the association of the MTR_Rex_MT with therapy response and of the MT_Rex_APT and APTw_asym_ with survival in this relatively small clinical cohort of 34 participants. Even though the contribution of mHb and Hs depositions to the contrast behavior of uncorrected and corrected tumor volumes could not be quantified, it seems reasonable to speculate that their amount was simply too small over the whole cohort to produce relevant effects. This might implicate that whilst larger mHb and Hs depositions are very visible, especially on MTR_Rex_APT contrast maps, advanced correction methods for the evaluation of CEST contrasts in the post therapeutic setting in representative clinical cohorts of patients with glioma could be of secondary relevance.

The relatively small cohort size, the subjective determination of mHb and Hs on T_1_w and T_2_*w imaging, and the lack of corresponding histopathological data for the assessed blood products are the major limitations of this study. Even though histopathological confirmation of remaining blood products in the tumor area is impossible to obtain, future studies assessing the impact of mHb and Hs on the clinical performance of CEST imaging in the early post-therapeutic interval might benefit from larger sample sizes (e.g., in multicenter trials) and support from AI-based automated segmentation tools for the definition of specific blood products.

## 5. Conclusions

A sophisticated correction for methemoglobin and hemosiderin did not substantially alter the clinical performance of APT and ssMT imaging at the first follow-up 4 to 6 weeks after the completion of radiotherapy in 34 participants with glioma. Larger blood product depositions were visible on APT and ssMT contrast maps and had minor effects on the clinical performance of the MTR_Rex_MT regarding therapy response assessment, and on that of the MTR_Rex_APT and APTw_asym_ regarding patient outcome prediction.

## Figures and Tables

**Figure 1 biomedicines-11-02348-f001:**
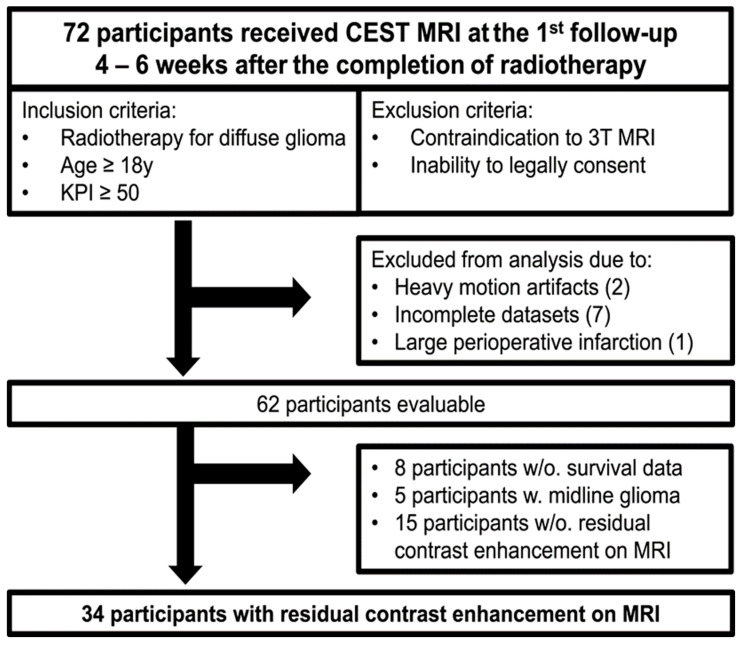
Flowchart. Displayed is a description of the whole study cohort participants and the 34 participants that were eventually included in the analysis. Between September 2018 and December 2021, 72 participants with diffuse glioma received CEST imaging 4 to 6 weeks after the completion of radiotherapy. In total, 11 participants had to be excluded from the analysis due to the indicated reasons. Given that diffuse midline gliomas are biologically distinct from hemispherical gliomas [27,28] and previous studies observed a dependency of CEST contrasts on the presence of residual tumor-associated contrast enhancement on MRI [6,11], the datasets of 34 participants with hemispherical gliomas, presence of residual contrast enhancement on MRI and available survival data were included in the analysis. The associations of mean CEST contrast values with therapy response and overall survival were tested by Kaplan–Meier analyses and log rank tests. KPI = Karnofsky Performance Score; w. = with; w/o. = without.

**Figure 2 biomedicines-11-02348-f002:**
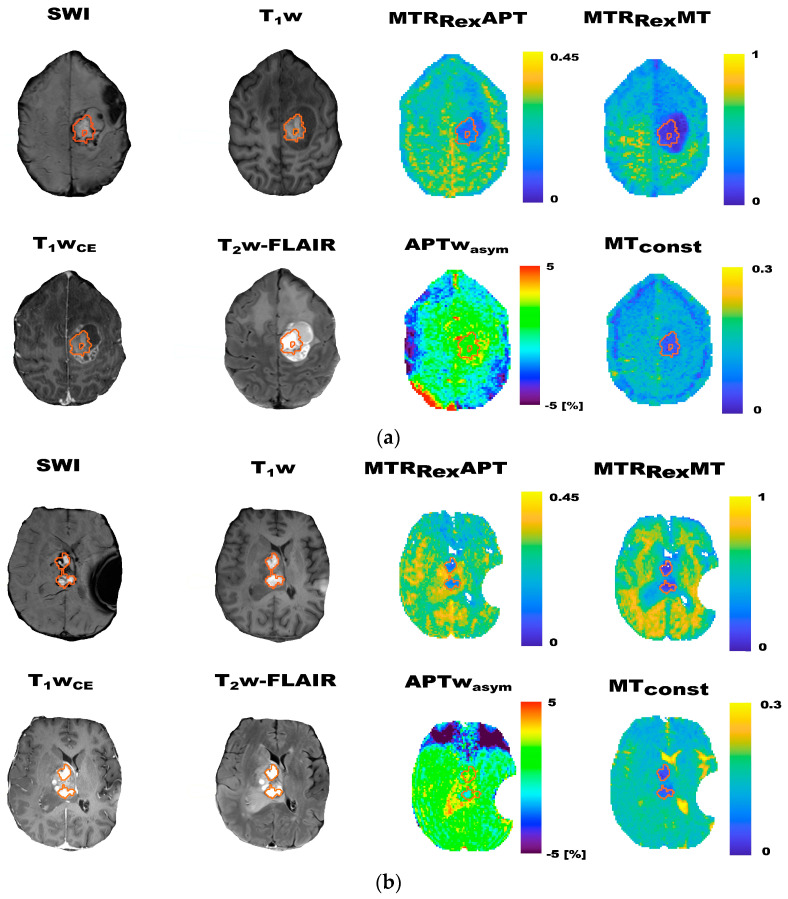
Exemplary contrast maps of two participants with larger methemoglobin depositions (mHb) in the tumor area (**a**,**b**). Given are T_2_* susceptibility-weighted (SWI), T_1_w, contrast-enhanced T_1_w (T_1_w_CE_) and T_2_w-FLAIR images, as well as CEST contrast maps of MTR_Rex_APT, MTR_Rex_MT, APTw_asym_ and MT_const_ imaging. The ROIs indicate the T_1_w-hyperintese mHb on all contrast maps. mHb visually showed markedly decreased values pronounced on MTR_Rex_APT, MTR_Rex_MT and MT_const_ contrast maps. The figure highlights the visible depression of the investigated CEST contrasts in correspondence to larger mHb depositions.

**Figure 3 biomedicines-11-02348-f003:**
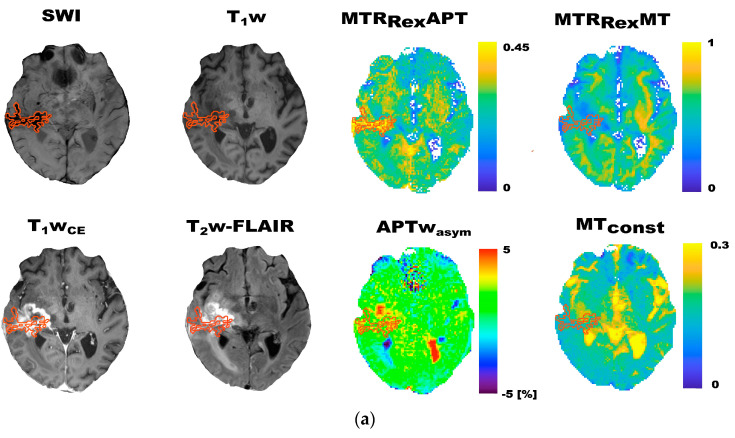
Exemplary contrast maps of two participants with larger hemosiderin depositions (Hs) in the tumor area (**a**,**b**). Given are T_2_* susceptibility-weighted (SWI), T_1_w, contrast-enhanced T_1_w (T_1_w_CE_) and T_2_w-FLAIR images, as well as CEST contrast maps of MTR_Rex_APT, MTR_Rex_MT, APTw_asym_ and MT_const_ imaging. The ROIs indicate Hs (dark on SWI) on all contrast maps. The MTR_Rex_APT visually showed markedly elevated contrast values corresponding sharply to Hs in these participants. The figure highlights the visible increase, especially of the MTR_Rex_APT in correspondence to larger Hs depositions.

**Figure 4 biomedicines-11-02348-f004:**
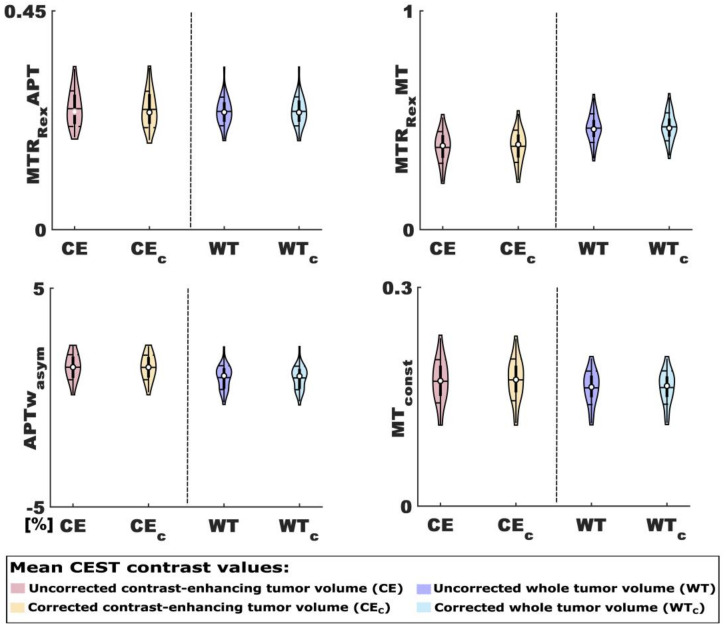
Mean CEST contrast values of uncorrected and corrected tumor volumes. Given are violin plots for MTR_Rex_APT, MTR_Rex_MT, APTw_asym_ and MT_const_ mean contrast values for contrast-enhancing and whole tumor volumes without (CE and WT) and with (CE_C_ and WT_C_) correction for mHb and Hs. The figure highlights that there were no relevant differences between the CEST contrast values of tumor volumes that were uncorrected and corrected for methemoglobin (mHb) and hemosiderin (Hs) depositions.

**Figure 5 biomedicines-11-02348-f005:**
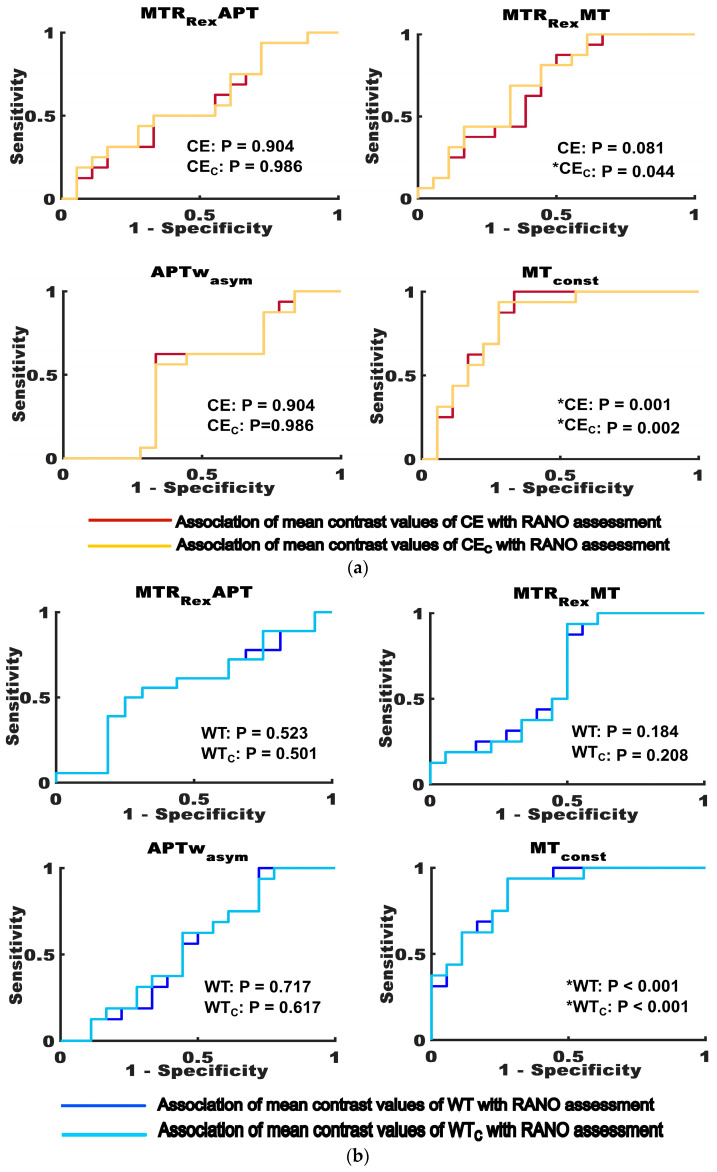
Association of mean CEST contrast values for uncorrected and corrected tumor volumes with therapy response. The figure shows the receiver operating characteristic (ROC) curves testing for the ability of MTR_Rex_APT, MTR_Rex_MT, APTw_asym_ and MT_const_ mean contrast values to differentiate between participants with progressive disease (higher mean values compared to the cohort median) and stable disease (lower mean values), as assessed according to the response assessment in neuro-oncology (RANO) criteria. (**a**) shows the results for uncorrected (CE—dark red) and corrected (CE_C_—yellow) contrast-enhancing tumor volumes. (**b**) shows the results for uncorrected (WT—dark blue) and corrected (WT_C—_light blue) whole tumor volumes. Statistically significant results (*p* ≤ 0.05) are indicated with an asterisk (*). The figure highlights that there were only marginal differences in the association of CEST contrast values with therapy response for uncorrected and corrected tumor volumes, which mainly affected the MTR_Rex_MT.

**Figure 6 biomedicines-11-02348-f006:**
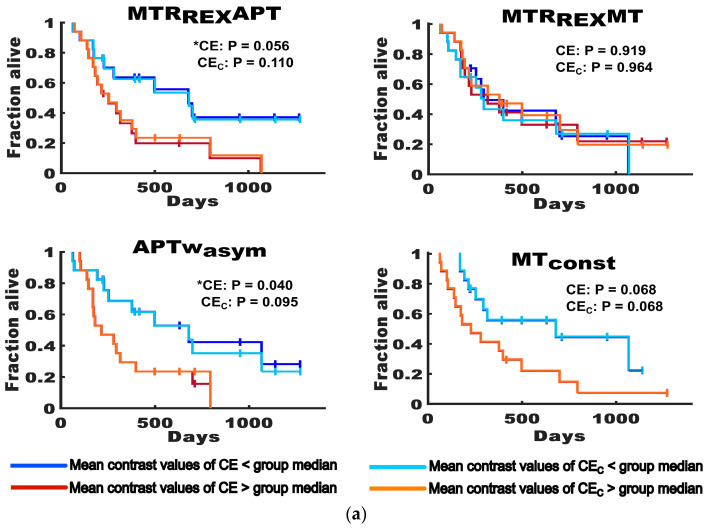
Association of the mean CEST contrast values for uncorrected and corrected tumor volumes with survival. The figure shows Kaplan–Meier plots displaying the association of MTR_Rex_APT, MTR_Rex_MT, APTw_asym_ and MT_const_ mean contrast values of uncorrected (CE and WT) and corrected (CE_C_ and WT_C_) tumor volumes with overall survival. (**a**) shows the plots for contrast-enhancing tumor volumes (CE and CE_C_). (**b**) shows the plots for whole tumor volumes (WT and WT_C_). The survival of participants with mean values below the group medians is indicated by dark blue graphs for uncorrected tumor volumes (CE and WT) and by turquoise graphs for corrected tumor volumes (CE_C_ and WT_C_). The survival of participants with mean values above the group medians is indicated by brown graphs for uncorrected tumor volumes (CE and WT) and by orange graphs for corrected tumor volumes (CE_C_ and WT_C_). Statistically significant results (*p* ≤ 0.05) are indicated with an asterisk (*). The figure highlights that there were only marginal differences in the association of CEST contrast values with overall survival that mainly affected the MTR_Rex_APT and the APTw_asym_.

**Table 1 biomedicines-11-02348-t001:** Clinical characteristics. Displays a summary of the most relevant clinical characteristics for all 34 evaluated study participants with glioma.

Characteristic		Number (n)	Percentage
Age at diagnosis	Mean 59.2 ± 15.6	34	
Therapy response at the 1st FU ^1^	Stable disease (SD)	16	47.1%
	Progressive disease (PD)	18	52.9%
Overall survival	Median 287 days (min. 63, max. 1271)		
Alive at data cut-off		10	29.4%
Sex	Male	19	55.9%
	Female	15	44.1%
Treatment for	Initial disease	31	91.2%
	Progressive disease	3	8.8%
Therapy	Radiation	6	17.6%
	Chemoradiation	28	82.4%
	Debulking surgery	21	61.8%
Diagnosis	GBM ^2^	28	82.4%
	Gliosarcoma	2	5.9%
	Astrocytoma	4	11.8%
WHO ^3^	II	1	2.9%
	III	1	2.9%
	IV	32	94.1%
*IDH* ^4^ status	*IDH*wt ^5^	28	82.4%
	*IDH*mut ^6^	4	11.8%
	n/a	2	5.9%
*MGMT* promotor methylation	Yes	19	55.9%
	No	12	35.3%
	n/a	3	8.8%

^1^ FU = follow-up MRI; ^2^ GBM = glioblastoma, ^3^ WHO II–IV = World Health Organization classification system for primary brain tumors grade II–IV; ^4^ *IDH* = isocitrate-dehydrogenase isotype 1/2; ^5^ wt = wildtype; ^6^ mut = mutation.

## Data Availability

Data generated or analyzed during the study are available from the corresponding author upon reasonable request.

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
