# Peer review of "Post-Surgical Depositions of Blood Products Are No Major Confounder for the Diagnostic and Prognostic Performance of CEST MRI in Patients with Glioma"

_biomedicines, 2023, doi:10.3390/biomedicines11092348_

Round 1

Reviewer 1 Report

The manuscript sounds technically average; however, I have following concerns should be addressed before any decision.  

1.      Please explain in your captions of figure and title of table, why are these tables or figures necessary in your paper? What are the purposes and what are the message you want to deliver via these figures and tables?

2.      The current metrics might not be sufficient to judge the performance of the model holistically. Please enhance the result analysis part of your paper.

3.      The existing literature should be classified and systematically reviewed, instead of being independently introduced one-by-one.

4.       In the introduction section, the motivations of the proposed access control model must be included in detail. The section numbering must be changed in the paper organization paragraph.

5.      The abstract is too general and not prepared objectively. It should briefly highlight the paper's novelty as what is the main problem, how has it been resolved and where the novelty lies?

6.      The 'conclusions' are a key component of the paper. It should complement the 'abstract' and normally used by experts to value the paper's engineering content. In general, it should sum up the most important outcomes of the paper. It should simply provide critical facts and figures achieved in this paper for supporting the claims.

7.      For better readability, the authors may expand the abbreviations at every first occurrence.

8.      The author should provide only relevant information related to this paper and reserve more space for the proposed framework.

Needs minor corrections

Author Response

We would very much like to thank the reviewer for his remarks and suggestions for improvement of the manuscript. We feel that the critical review helped to improve the overall quality of this manuscript. The individual concerns are addressed point-by-point below.

Answers to the Reviewer are written in blue color. Changes to the manuscript are written in red color. Reviewer’s comments are written in black color.

Point-by-point response to Reviewer 1:

  1. Please explain in your captions of figure and title of table, why are these tables or figures necessary in your paper? What are the purposes and what are the message you want to deliver via these figures and tables?

R1.1)

  • Thank you very much for this question and suggestions. The tables and figures display the results of our analyses and are intended to underscore our interpretation of the data in the following way. Explanatory sentences have been introduced as indicated:
    1. Figure 1 is intended to give an overview of the whole study cohort and the 34 participants that have been included in the analysis, after exclusion of 38 participants due to the reasons outlined in the text. An explanatory sentence has been introduced in the legend: “…..Displayed is a description of the whole study cohort participants and the 34 participants that were eventually included in the analysis….”
    2. Table 1 summarizes the outcomes of the clinical assessments that were been performed in this analysis, as well as the most relevant clinical characteristics that are known to be associated with the occurrence of gliomas and their clinical course. An explanatory sentence has been introduced in the head line: “…. Displays a summary of the most relevant clinical characteristics for all 34 evaluated study participants with glioma….”
    3. Figure 2 shows anatomical images and CEST contrast maps for an exemplary participant with larger methemoglobin (mHb) depositions. The figure is intended to visualize the remarkable drop of CEST-contrast values in correspondence to mHb. An explanatory sentence has been introduced in the figure legend: “….The figure highlights the visible depression of the investigated CEST contrasts in correspondence to larger mHb depositions…”
    4. Figure 3 shows anatomical images and CEST contrast maps for an exemplary participant with larger hemosiderin (Hs) depositions. The figure is intended to visualize the remarkable increase of MTRRexAPT contrast values in correspondence to Hs. An explanatory sentence has been introduced in the figure legend: “….The figure highlights the visible increase especially of the MTRRexAPT in correspondence to larger Hs depositions…”
    5. Figure 4 is intended to visualize that there were no relevant differences in the mean CEST contrast values between uncorrected and corrected tumor volumes. An explanatory sentence has been introduced in the figure legend: “…The figure highlights, that there were no relevant differences between the CEST contrast values of tumor volumes that were uncorrected and corrected for methemoglobin (mHb) and hemosiderin (Hs) depositions…”
    6. Figure 5 is intended to highlight that there were only marginal differences in the association of the CEST contrast values of uncorrected and corrected tumor volumes with therapy response. An explanatory sentence has been introduced in the figure legend: “….The figure highlights that there were only marginal differences in the association of CEST contrast values with therapy response for uncorrected and corrected tumor volumes, which mainly affected the MTRRexMT….”
    7. Figure 6 is intended to highlight that there were only marginal differences in the association of the mean CEST contrast values of uncorrected and corrected tumor volumes with survival. An explanatory sentence has been introduced in the figure legend: “…..The figure highlights that there were only marginal differences in the association of CEST contrast values with overall survival that mainly affected the MTRRexAPT and the APTwasym…”
    8. Figure A1 is intended to illustrate the blood product and tumor volumes, as well as the differences between uncorrected and corrected tumor volumes on anatomical imaging. An explanatory sentence has been introduced in the figure legend: “….The figure illustrates how the investigated bloodproducts and uncorrected tumor volumes were defined on T1w, SWI, T1wCE and T2w-FLAIR imaging in an exemplary participant with coexisting depositions of mHb and Hs and highlights the differences between uncorrected and corrected tumor volumes….”
    9. Figure A2 is intended to illustrate the T1 contributions of mHb and Hs depositions that affect CEST the investigated CEST contrasts. An explanatory sentence has been introduced in the legend: “…The figure highlights the T1 contribution of mHb and Hs depositions that affact especially MTconst, but also MTRRexAPT and MTRRexMT imaging…”
    10. Table A1 quantifies the CEST contrast values of metheoglobin (mHb), hemosiderin (Hs) and corrected (CEC and WTC) tumor volumes. The intention is to quantify the visible characteristics of the CEST contrasts in correspondence to mHb and Hs and compare them to the CEST contrast behavior of corrected tumor volumes. Whilst the depression of CEST contrast values in correspondence to mHb could be quantified, the visibly increased MTRRexAPT values were not matched by numeric findings. An explanation has been introduced in the legend: “….The table quantifies the lower mean MTRRexAPT-, MTRRexMT- and MTconst – values for mHb in comparison to corrected tumor volumes that is visually demonstrated for an examplary participant in Figure 2. Allthough very well visible on Figure 3, the MTRRexAPT did not show higher mean values for Hs in comparison to corrected tumor volumes…”
    11. Table A2 and A3 just summarize the results of the ROC and Kaplan-Meyer-Analysis given in section 3.3 and 3.4 and displayed in Figures 5 and 6.

  1. The current metrics might not be sufficient to judge the performance of the model holistically. Please enhance the result analysis part of your paper.

R1.2) Thank you very much for this consideration. The current metrics have been previously used to assess the association of the investigated CEST contrasts with therapy response and survival. The results of these analyses have been published in the peer-reviewed journals “Radiotherapy and Oncology” (DOI: 10.1016/j.radonc.2023.109694) and “Magnetic Resoncance in Medicine” (DOI: 10.1002/mrm.29746). Since larger depositions of blood products are very well visible on CEST imaging but not readily distinguishable on anatomical imaging in the case of hemosiderin, the purpose of this study was to assess, whether clinical studies investigating CEST contrasts in patients with brain tumors in a postsurgical setting should include advanced correction methods for blood break down products.

  1. The existing literature should be classified and systematically reviewed, instead of being independently introduced one-by-one.

R1.3) Thank you very much for this consideration. The literature references have been re-checked. They encompass the most relevant studies investigating the association of CEST imaging of the APT and ssMT with therapy response and survival of patients with glioma before and after radiotherapy, the most relevant studies investigating the ability of APT and ssMT imaging to differentiate radiation necrosis and glioma progression after radiotherapy, the studies describing the relevant technical parameters of the investigated CEST contrasts, and the existing studies investigating the CEST contrast behaviours of blood products. These studies have been introduced in a systemic fashion throughout the paper at the appropriate text passages. Since this study is an original research paper investigating the influence of postsurgical blood products on the clinical performance of APT and ssMT CEST imaging in the early post radiotherapy interval and not a metanalysis, we hope that reviewer follows our rational.

  1. In the introduction section, the motivations of the proposed access control model must be included in detail. The section numbering must be changed in the paper organization paragraph.

R1.4) We would like to thank the reviewer for his consideration. This issue has been rechecked to ensure that the research question is clearly expressed and the chosen technical set-up explained in sufficient detail. The section numbering is comprehensive and in line with the style guide.

  1. The abstract is too general and not prepared objectively. It should briefly highlight the paper's novelty as what is the main problem, how has it been resolved and where the novelty lies?

R1.5) Thank you very much for this consideration. The abstract has been re-checked to make sure that it briefly but clearly highlights the research question, explains the study set-up, outlines the results and addresses the conclusion that was drawn from these results. We feel that the abstract is concise and clearly indicates the novelty and relevance of this study: Investigation of the impact of postsurgical blood-break-down-products on the clinical performance of CST imaging in patients with glioma in the early post radiotherapy interval à Clinical studies investigating CEST imaging in the early poster radiotherapy interval do not necessarily need advanced correction methods for remaining post surgical blood break down products, even though methemoglobin and hemosiderin depositions are very well visible on CEST contrast maps.

  1. The 'conclusions' are a key component of the paper. It should complement the 'abstract' and normally used by experts to value the paper's engineering content. In general, it should sum up the most important outcomes of the paper. It should simply provide critical facts and figures achieved in this paper for supporting the claims.

R1.6) Thank you very much for this conserideration. The conclusion of our manuscript has been re-checked to make sure that is in line with the raised points of concern.

  1. For better readability, the authors may expand the abbreviations at every first occurrence.

R1.7) Thank you very much for this concern. The introduction of the abbreviations at the first use throughout the text and figure legends has been re-checked verified to ensure convenient readability of the text.

  1. The author should provide only relevant information related to this paper and reserve more space for the proposed framework.

R1.8) Thank you very much for this concern. This concern has been checked and we feel that we only provide the information that is relevant to understand our analysis and conclusion.

Reviewer 2 Report

I have the following comments:

-The Abstract would better be written in a structured form for better clarity, e.g., subdivided into "Background", "Materials and methods", "Results" and "Conclusions" sections. Especially the study conclusions should be stressed at the end of the Abstract - in this context, the word "However" at line 34 should be removed.

-Introduction. A brief (one sentence) explanation of the basics of CEST imaging should be provided for clarity of readers without a specific expertise on the topic.

-Materials and methods. How was a number of 72 participants decided for this prospective study? Was an apriori power analysis performed to establish a minimum sample numerosity? And could the unbalanced patient distribution (61 with initial disease and 11 with relapsing/progressive disease) have introduced some bias in the results?

-Materials and methods, line 89. What did you mean by "excessive" perioperative ischemia?

-Results. Please avoid subtitles (e.g., "3.1. Visually decreased MTRRexAPT, MTRRexMT and ... maps) summarizing the main findings illustrated in each subsection. Such subtitles should introduce the topic instead, e.g. "MTRRexAPT, MTRRexMT and MTconst values", "Impact of correction on tumor mean CEST contrast values" and so on.

Author Response

Post-surgical depositions of blood products are no major confounder for the diagnostic and prognostic performance of CEST MRI in patients with glioma

We would very much like to thank the reviewer for his comments and suggestions which helped to improve the manuscript. The raised concerns are addressed point-by-point in the following.

Answers to the Reviewer are written in blue color. Changes to the manuscript are written in red color. Reviewer’s comments are written in black color.

Point-by-point response to Reviewer 2:

  1. The Abstract would better be written in a structured form for better clarity, e.g., subdivided into "Background", "Materials and methods", "Results" and "Conclusions" sections. Especially the study conclusions should be stressed at the end of the Abstract - in this context, the word "However" at line 34 should be removed.

R2.1) Thank you very much for this consideration. In accordance with the journal style guidelines the abstract is written in a single paragraph in a structured form but without headings. The word “however” at the end of the abstract, which was intended to highlight the greater conclusion in the abstract format required by the journal, has been removed.

  1. A brief (one sentence) explanation of the basics of CEST imaging should be provided for clarity of readers without a specific expertise on the topic.

R2.2) Thank you very much for this suggestion. A short explanation on the basics of CEST imaging has been introduced in the introduction: “... Imaging of the amide-proton transfer (APT) and of the semi-solid magnetization transfer (ssMT) relies on the selective radio frequency (RF) saturation of protons bound in peptide bindings or sub-cellular macromolecules, with subsequent magnetization transfer to bulk water through chemical exchange or spin-spin couplings respectively [5] (collec-tively referred to as CEST imaging in the following)…”

  1. Materials and methods. How was a number of 72 participants decided for this prospective study? Was an apriori power analysis performed to establish a minimum sample numerosity?

R2.3.i) All patients who were treated for diffuse glioma at the Department of Radiation Oncology of the University Hospital Heidelberg between September 2018 and December 2021 with an age of 18 years or older, a Karnovsky-Performance-Score of 50 or higher and with the ability to consent were eligible for study inclusion. A power analysis was not performed in advance to calculate the needed number of study participants. The recruitment process had to be paused for large parts of 2019 due to the break of a major water pipe in the building of the MR scanner and was again intensified from October 2020 onwards. At the cut-off date on 31. of December 2021 72 participants had been included in the study.

And could the unbalanced patient distribution (61 with initial disease and 11 with relapsing/progressive disease) have introduced some bias in the results?

R2.3.ii) The unbalanced distribution of patients with initial and relapsing disease should not have introduced any bias into the observed results, since we investigated the impact of advanced correction methods for blood products on the clinical performance of the analyzed CEST-contrasts in a representative clinical cohort, regardless of primary tumor manifestation or progressive disease.

  1. Materials and methods, line 89. What did you mean by "excessive" perioperative ischemia?

R2.4) minor to moderate perioperative ischemia is very frequent in patients with glioma and results from unavoidable compression of brain tissue during the resection process. However this participant had perioperative infarction of large parts of the occipital lobe possibly due to injury of a supplying branch of the posterior cerebral artery or inadequate perioperative compression of the brain tissue. Ischemia is usually followed by a disruption of the blood-brain-barrier a couple of days to weeks after the adverse event, which results in hemorrhagic transformation of the infarcted area. Since the affected area exceeded the margins of the tumor, this case needed to be excluded from the analysis.

  1. Please avoid subtitles (e.g., "3.1. Visually decreased MTRRexAPT, MTRRexMT and ... maps) summarizing the main findings illustrated in each subsection. Such subtitles should introduce the topic instead, e.g. "MTRRexAPT, MTRRexMT and MTconst values", "Impact of correction on tumor mean CEST contrast values" and so on.

R2.5) Thank you very much for this consideration. The subtitles have been adapted accordingly:

  • 1. CEST contrast maps of participants with larger depostions of mHb and Hs
  • 2. Differences between the mean CEST contrast values of uncorrected and corrected tumor volumes
  • 3. Association of CEST contrast values of uncorrected and corrected tumor volumes with therapy response
  • 4. Association of mean CEST contrast values of uncorrected and corrected tumor volumes with overall survival
  • 5. Supermedian analysis of the mean CEST contrast values of uncorrected and corrected tumor volumes

Round 2

Reviewer 1 Report

No further comments.

No further comments.

Author Response

Thank you very much, for the second review. The Reviewer has left no further comments.

Reviewer 2 Report

Thank you. No further comments.

Author Response

Thank you very much for the second review. The reviewer has left no further comments.